# *OsGGC2*, Gγ Subunit of Heterotrimeric G Protein, Regulates Plant Height by Functionally Overlapping with *DEP1* in Rice

**DOI:** 10.3390/plants11030422

**Published:** 2022-02-03

**Authors:** Genki Chaya, Shuhei Segami, Moeka Fujita, Yoichi Morinaka, Yukimoto Iwasaki, Kotaro Miura

**Affiliations:** 1Faculty of Bioscience and Biotechnology, Fukui Prefectural University, 4-1-1 Kenjojima, Matsuoka, Yoshida-gun, Eiheiji-cho 910-1195, Fukui, Japan; s2093004@g.fpu.ac.jp (G.C.); pt-fujita@fpu.ac.jp (M.F.); morinaka@fpu.ac.jp (Y.M.); iwasaki@fpu.ac.jp (Y.I.); 2Research Institute of Environment, Agriculture and Fisheries, 442 Shakudo, Habikino 583-0862, Osaka, Japan; segamis@mbox.kannousuiken-osaka.or.jp

**Keywords:** rice, heterotrimeric G protein, *DEP1*, *OsGGC2*

## Abstract

Plant heterotrimeric G proteins have been shown to regulate the size of various organs. There are three types of Gγ subunits in plants: type A, consisting of a canonical Gγ domain; type B, possessing a plant-specific domain at the N-terminus of the Gγ domain; and type C, possessing a plant-specific domain at the C-terminal of the Gγ domain. There is one type A, one type B, and three type C of the five γ-subunits in the rice genome. In type C Gγ subunits, *GS3*, which controls grain size; *DEP1*, which controls plant height and panicle branching; and their homolog *OsGGC2*, which affects grain size, have been reported; however, the function of each gene, their interactions, and molecular mechanisms for the control of plant height have not yet been clarified. In this study, we generated loss-of-function mutants of *DEP1* and *OsGGC2*, which have high homology and similar expression, and investigated their phenotypes. Since both *dep1* and *osggc2* mutants were dwarfed and the double mutants showed a synergistic phenotype, we concluded that both *DEP1* and *OsGGC2* are positive regulators of plant height and that their functions are redundant.

## 1. Introduction

Heterotrimeric G proteins consist of three subunits (Gα, Gβ, and Gγ) in mammals and yeast cells. They act as signal transducers by transferring extracellular information to intracellular components [1,2,3,4]. External signals bind or affect G protein-coupled receptors (GPCRs) to activate them. Activated GPCRs, which function as intrinsic GDP/GTP exchange factors, convert Gα-GDP to Gα-GTP. When GTP binds to the α-subunit (Gα-GTP), heterotrimeric G proteins dissociate into Gα-GTP and Gβγ dimer. Gα-GTP and the Gβγ dimer can regulate the respective effector molecules. Although plants have no cognate GPCRs with nucleotide exchange activity [5], heterotrimeric G proteins also regulate organ development and transmit external signals to intracellular target proteins in plants [6,7,8].

In plants, there are two types of Gα subunits: canonical Gα subunits that are structurally similar to their animal counterparts, and unconventional extra-large Gα subunits (XLGs); the latter contain a C-terminal domain homologous to the canonical Gα along with an extended N-terminal domain [9]. There is one Gα and four putative XLGs in the rice genome [10,11,12]. The loss-of-function mutant of rice Gα, *dwarf1* (*d1*), shows a distinct dwarf, shortened panicle, and short and round grain phenotypes [10]. All four XLG mutants showed early heading and higher salinity tolerance, of which *xlg1* and *xlg3* exhibited slightly dwarf phenotypes [12].

There is one canonical Gβ in the rice genome (*RGB1*), whose RNAi causes severe phenotypes resulting in seedling lethality and mild phenotypes resulting in dwarfing, grain shortening, and browning of the lamina joint regions and nodes. The observations of the *d1 rgb1* double mutant suggest that these *rgb1* phenotypes are regulated by different pathways from that of Gα [13].

There are three types of Gγ subunits in plants: type A, consisting of a canonical Gγ domain; type B, possessing a plant-specific domain at the N-terminus of the Gγ domain; and type C, possessing a plant-specific, long, cysteine-rich domain at the C-terminus of the Gγ domain [14]. The composition of γ-subunits is diverse among plant species: *Arabidopsis* has two of type A and one of type C, whereas rice has one of type A (*RGG1*/Gγ1), one of type B (*RGG2*/Gγ2), and three of type C (*GS3*, *DEP1*, and *OsGGC2*) of the five γ subunits [15].

An in-frame mutation in the γ domain of Gγ2 resulted in a larger grain size [16]. *GS3* and *DEP1* have been cloned as genes that control grain length and panicle branching and length, respectively, and have been shown to affect agronomic traits [17,18]. Subsequently, GS3 and DEP1 are homologous to the Gγ subunit, and OsGGC2 is homologous to both [15]. GS3 has been shown to result in longer grains in loss-of-function mutants and incompletely dominant short grains in C-terminal deletions, indicating that GS3 function is limited to grain size control [17,19]. It has been reported that loss of the cysteine-rich domain of DEP1 decreases plant height and panicle length and increases panicle branching [18]. The expression pattern of *OsGGC2* is similar to that of *DEP1*—loss of its function has been shown to slightly shorten the grain, and its overexpression leads to longer grains [20].

In this study, to understand the function of *OsGGC2* and its redundancy with *DEP1*, we generated loss-of-function mutants of *OsGGC2*, *DEP1,* and their double mutants.

## 2. Results

### 2.1. Loss-of-Function Mutants of DEP1 and OsGGC2

To observe the phenotypes in detail, we generated loss-of-function mutants of *DEP1* and *OsGGC2* using the CRISPR-Cas9 system [21]. We sequenced the transformants and found that we obtained a *dep1* mutant with a 7 bp deletion at 145 bp to 151 bp, which resulted in a premature stop codon at 62 amino acids (Figure 1a,c). The *osggc2* mutant had a 1 bp insertion at 96 bp, resulting in a premature stop codon at 44 amino acids (Figure 1b,d). Both mutations occurred at the beginning of the conserved Gγ domain and produced very short, truncated proteins; therefore, we concluded that these were loss-of-function mutants.

### 2.2. dep1 and osggc2 Show a Similar Semi-Dwarf Phenotype

Plant observations revealed that both *dep1* and *osggc2* showed a semi-dwarf phenotype. The plant height of *dep1* and *osggc2* was 8% shorter than that of the wild type (WT) (Figure 2a,b,e). The panicle length of *dep1* and *osggc2* was not significantly different from that of the WT (Figure 2c,f), and the grain length was slightly shorter only in *dep1* (Figure 2d,g). These results indicate that *DEP1* and *OsGGC2* are positive regulators of plant height, and *DEP1* is a positive regulator of grain length.

### 2.3. d1 Is Almost Epistatic to the Dwarf Phenotypes of dep1 and osggc2

To clarify the signal transduction of heterotrimeric G proteins in rice, we observed the phenotypes *d1*
*dep1* and *d1*
*osggc2* double mutants that were created by crossing the *dep1* and *osggc2* mutants with the Gα loss-of-function mutant *d1*. Although both double mutants showed a dwarf phenotype similar to that of *d1*, the plant heights of *d1 dep1* and *d1 osggc2* were slightly shorter than that of *d1* (Figure 3a,b,e). Panicle lengths of *d1 dep1* and *d1 osggc2* were also slightly shorter than in *d1* (Figure 3c,f). The grain lengths of *d1 dep1* and *d1 osggc2* were not significantly different from *d1* (Figure 3d,g). These results suggest that *d1* is almost epistatic to *dep1* and *osggc2* in plant height, although there was also a slight additive phenotype. Therefore, DEP1 and OsGGC2 regulate plant height and panicle length mainly by Gα-mediated signaling pathways and partially by other pathways. The single mutant of *dep1* produced shorter grains (Figure 2d,g), and *d1 dep1* had the same grain length as *d1* (Figure 3d,g), indicating that *d1* is epistatic to *dep1* for grain length.

### 2.4. OsGGC2 and DEP1 Redundantly Regulate Plant Height and Panicle Length in Rice

Given that DEP1 and OsGGC2 have 66% protein similarity and show similar gene expression patterns during the vegetative and reproductive phases [20], we speculated that there is functional redundancy. We produced a double mutant, *dep1 osggc2*, by crossing the mutants to investigate this hypothesis. Interestingly, the plant height of *dep1 osggc2* was significantly shorter than that of its parents and *d1* mutants (Figure 2e and Figure 4a,b,e). The panicle length of *dep1 osggc2* was also shorter than that of the parents and *d1* (Figure 2f and Figure 4c,f). The grain size of *dep1 osggc2* was slightly smaller than that of the WT but longer than that of the *d1* mutant (Figure 4d,g). Since the *dep1 osggc2* plant had a synergistic effect on plant height and panicle length, we concluded that DEP1 and OsGGC2 have redundant functions in plant height and panicle length regulation.

## 3. Discussion

On the basis of these results, we conclude that DEP1 and OsGGC2 act redundantly to regulate plant height through a Gα-mediated pathway and through another pathway, and that DEP1 regulates grain length through the Gα-mediated pathway, but OsGGC2 does not. In a previous study [20], the grain length of the knockout mutant of *GGC2* was slightly shorter in the genetic background of ZH11, but our results contradict this report. This discrepancy in the results indicates that differences in the genetic background affect the control of grain length by the heterotrimeric G protein pathway and suggest the presence of unknown genes that affect grain length regulation by the G protein pathway. To date, more than 40 genes controlling grain length have been reported in rice [22]. By clarifying the relationship between each of these genes and G protein-related genes, we can elucidate the whole picture of plant G protein signals.

What are the possible signaling pathways not mediated by Gα? A more severe dwarf phenotype than *d1*, similar to the *dep1 osggc2* mutant, has been shown in the *tud1* mutant [23]. *TUD1* encodes a U-box E3 ubiquitin ligase, and the *tud1* mutation is epistatic to *d1.* TUD1 has been reported to be involved in the brassinosteroid (BR) signaling pathway, and DEP1 and OsGGC2 may also be related to the TUD1 and/or BR signaling pathways.

Among the four XLGs in rice, *xlg1* and *xlg3* mutants show a slightly dwarf phenotype [12]. It is possible that XLG1 and XLG3 may have signaling pathways that regulate plant height as interaction targets of DEP1 and OsGGC2.

In addition, RNAi mutants of Gβ display browning of the lamina joint and dwarf phenotypes, which are also present in the *d1* background. Thus, there may be two signaling pathways in rice: one directly controlled by Gα and the other by the Gβγ dimer.

This study revealed that two type C Gγ subunits, DEP1 and OsGGC2, are positive regulators and are functionally redundant for plant height in rice. Another type C Gγ, GS3, clearly functions only in grains, indicating that the three Gγ subunits are functionally differentiated in rice.

In *Arabidopsis*, there is only one type C Gγ subunit, but its copy number and function vary among different species. Two type C Gγ subunits positively regulate plant height, and one type C Gγ subunit negatively regulates grain length in rice. This suggests that, evolutionarily, taller plant height was essential for survival, and it was necessary to control plant height and grain size separately. In addition, loss-of-function mutants of *GS3* and loss-of-C-terminus mutants of *DEP1* have been used as high-yielding genes that increase grain size, semi-dwarfism, and grain number. This suggests that the G protein pathway is an essential target for crop breeding, and it is potentially an important breeding target for organ size in other crops.

## 4. Materials and Methods

### 4.1. Plant Materials and Growth Conditions

A rice cultivar, Nipponbare (*Oryza sativa* L. ssp. *japonica*), was used as the WT and for targeted mutagenesis. The mutant DK22 in Nipponbare background was used as the *d1* mutant [10]. All plants were grown in 10 × 10 cm pots in a closed growth chamber (TGH; ESPEC MIC, Osaka, Japan) at 28 °C from 07:00 to 21:00 and 25 °C from 21:00 to 07:00 for three months and then at 28 °C from 09:00 to 19:00 and 25 °C from 19:00 to 09:00 for the next two months.

### 4.2. Production of dep1 and osggc2 Mutants

The mutants were produced using the CRISPR/Cas9 system developed by Mikami et al. [21]. The primer sets, dep1-CRISPR-oligo-U: GATGAGCTTCACTTCCTTGA and dep1-CRISPR-oligo-L: TCAAGGAAGTGAAGCTCATC, and osggc2-CRISPR-oligo-U: CAGATCCTCAACCGGGAGGT and osggc2-CRISPR-oligo-L: ACCTCCCGGTTGAGGATCTG, were annealed and cloned into pU6gRNA as the target sequence. The target sequence with the OsU6 promoter was replaced with the pZH_gYSA_MMCas9 vector. The construct was transformed into Nipponbare calli as described by Mikami et al. [20]. Thereafter, regenerated M_0_ plants were transplanted and sequenced using a 3130xl Genetic Analyzer (Thermo Fisher Scientific, Waltham, MA, USA) with the following primer sets: dep1-SQ-U: ACTGGTCAATTACTCCAGGC and dep1-SQ-L: GGGCCTAAGTGTGACATACA, and osggc2-SQ-U: TTAGCTACGCAGCTGTAGCC and osggc2-SQ-L: CACTGTGAAAGAAAGCGAGG. Seedlings generated from M_1_ seeds were genotyped via sequencing using the above primer sets, and homozygous plants were selected.

### 4.3. Production of Double Mutants

Double mutants were produced by crossing two parental mutants. After the germination of F2 seeds obtained from F1 plants, the genotypes of the seedlings were determined by sequencing. The homozygous plants of *dep1* and *osggc2* were selected using the primers described above, and *d1* was selected using the primer set DK22-SQ-U: TTCAGGTGAAAACAAATAGCC and DK22-SQ-L: CCTTCGTTATGTAGACTGCG.

### 4.4. Phenotype Evaluation

Plant height was measured from the bottom to the top of the main column, and panicle length was measured from the bottom to the top of the panicle of the main column; the means of five plants from each genotype were compared. Grain length was measured for 15 grains per plant, and the means of five plants from each genotype were compared.

### 4.5. DNA and Amino Acid Sequences

The DNA and amino acid sequences used in this study were obtained from the public data of RAP-DB (https://rapdb.dna.affrc.go.jp/, accessed on 10 January 2022). The locus IDs of *DEP1* and *OsGGC2* are Os09t0441900 and Os08g0456600, respectively.

## Figures and Tables

**Figure 1 plants-11-00422-f001:**
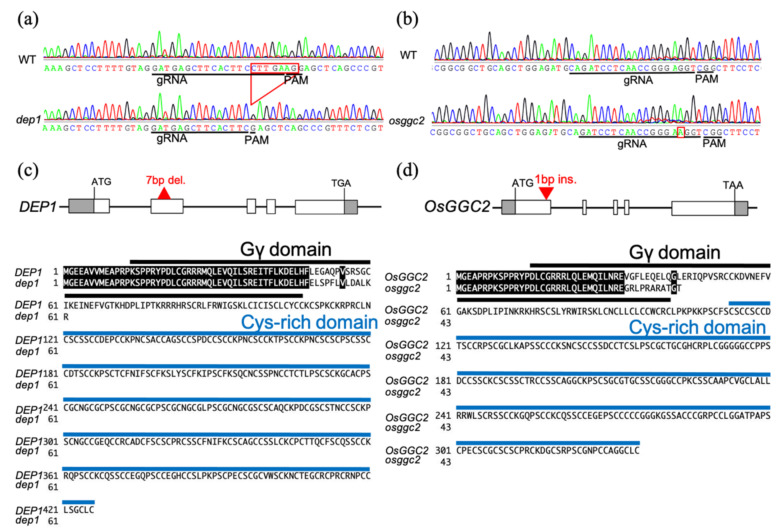
Mutation of *dep1* and *osggc2*. (**a**) The results of DNA sequencing of *dep1* mutant. Red box indicates 7 bp deletion of *dep1*. (**b**) The results of DNA sequencing of *osggc2* mutant. Red box indicates 1 bp insertion of *osggc2*. (**c**) Gene structure and amino acid sequence of *DEP1*. White and grey boxes indicate the coding and untranslated regions of the exon, respectively. Black and blue lines indicate Gγ domain and cysteine-rich domain, respectively. Unchanged amino acids in the mutants are indicated by black boxes. (**d**) Gene structure and amino acid sequence of *OsGGC2*. White and grey boxes indicate the coding and untranslated regions of the exon, respectively. Black and blue lines indicate Gγ domain and cysteine-rich domain, respectively. Unchanged amino acids in the mutants are indicated by black boxes.

**Figure 2 plants-11-00422-f002:**
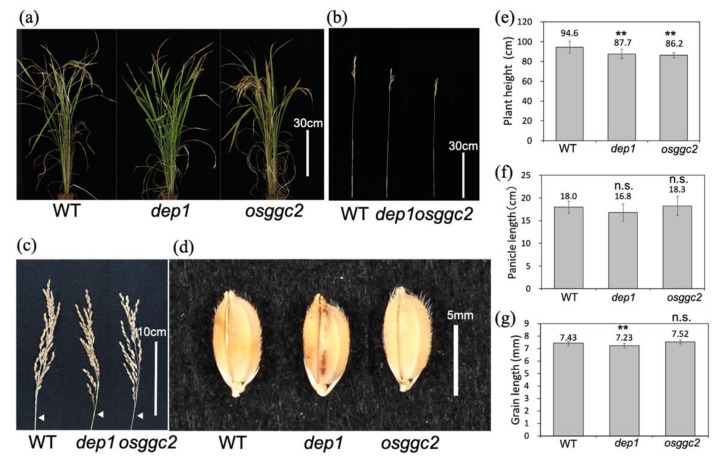
Phenotypes of *dep1* and *osggc2*. (**a**) Gross morphology of *dep1* and *osggc2*. Bar indicates 30 cm. (**b**) Main culm morphology of *dep1* and *osggc2*. Bar indicates 30 cm. (**c**) Panicle morphology of *dep1* and *osggc2*. Bar indicates 10 cm. White arrows indicate panicle bases. (**d**) Grain morphology of *dep1* and *osggc2*. Bar indicates 5 mm. (**e**) Graph of plant height of *dep1* and *osggc2*. (**f**) Graph of panicle length of *dep1* and *osggc2*. (**g**) Graph of grain length of *dep1* and *osggc2*. Student’s *t*-tests were performed between WT and mutants are indicated in (**e**–**g**). ** *p* < 0.01. n.s.: not significant. *n* = 5. Error bars indicate SD.

**Figure 3 plants-11-00422-f003:**
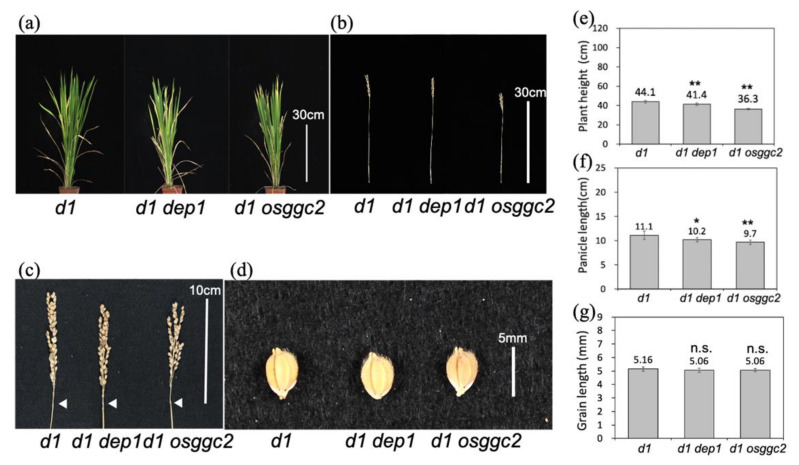
Phenotypes of double mutants *d1 dep1* and *d1 osggc2*. (**a**) Gross morphology of *d1 dep1* and *d1 osggc2*. Bar indicates 30 cm. (**b**) Main culm morphology of *d1 dep1* and *d1 osggc2*. Bar indicates 30 cm. (**c**) Panicle morphology of *d1 dep1* and *d1 osggc2*. Bar indicates 10 cm. White arrows indicate panicle bases. (**d**) Grain morphology of *d1 dep1* and *d1 osggc2*. Bar indicates 5 mm. (**e**) Graph of plant height of *d1 dep1* and *d1 osggc2*. (**f**) Graph of panicle length of *d1 dep1* and *d1 osggc2*. (**g**) Graph of grain length of *d1 dep1* and *d1 osggc2*. Student’s *t*-tests performed between the *d1* and double mutants are indicated in (**e**–**g**). ** *p* < 0.01. * *p* < 0.05. n.s.: not significant. *n* = 5. Error bars indicate standard deviation.

**Figure 4 plants-11-00422-f004:**
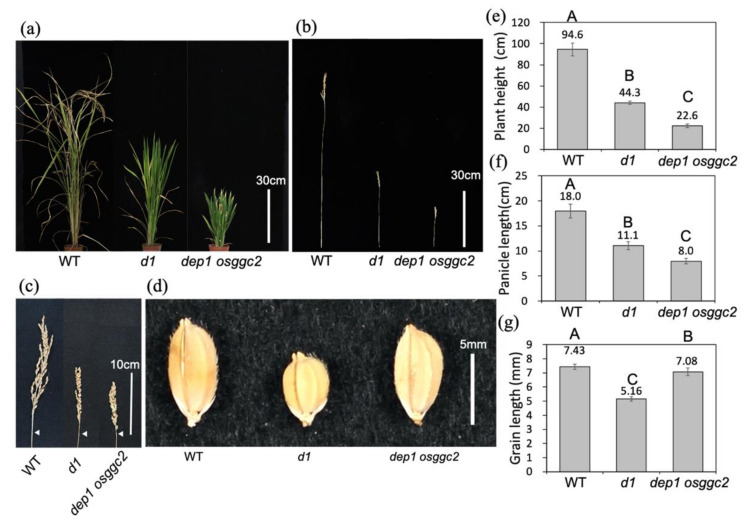
Phenotype of double mutant *dep1 osggc2*. (**a**) Gross morphology of *dep1 osggc2*. Bar indicates 30 cm. (**b**) Main culm morphology of *dep1 osggc2*. Bar indicates 30 cm. (**c**) Panicle morphology of *dep1 osggc2*. Bar indicates 10 cm. White arrows indicate panicle bases. (**d**) Grain morphology of *dep1 osggc2*. Bar indicates 5 mm. (**e**) Graph of plant height of *dep1 osggc2*. (**f**) Graph of panicle length of *dep1 osggc2*. (**g**) Graph of grain length of *dep1 osggc2*. Least significant difference tests of the wild type (WT) and the mutant are indicated in (**e**–**g**). Different letters indicate significant differences (*p* < 0.05). *n* = 5. Error bars indicate standard deviation.

## Data Availability

Not applicable.

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
