# Peer review of "OsGGC2, Gγ Subunit of Heterotrimeric G Protein, Regulates Plant Height by Functionally Overlapping with DEP1 in Rice"

_plants, 2022, doi:10.3390/plants11030422_

Round 1
Reviewer 1 Report
The paper “OsGGC2, Gγ subunit of heterotrimeric G protein, regulates plant height by functionally overlapping with DEP1 in rice” reported the genetic mechanisms of Gγ subunit of heterotrimeric G protein, OsGGC2, for plant height. Authors generated loss-of-function mutants of DEP1 and OsGGC2 in rice and found that both dep1 and osggc2 mutants showed dwarf phenotype and double mutants dep1 osggc2 also showed a synergistic phenotype. Authors concluded that both DEP1 and OsGGC2 are positive regulators of plant height, and their functions are redundant.
This is a technically competent study that provides strong evidence for two type C Gγ subunits, DEP1 and OsGGC2, are positive regulators and functionally redundant for plant height in rice.
However, there are some points to check and revise before publishing.
1. In 4.2, authors mentioned that “M1 seeds were genotyped by sequencing, and homozygous plants were selected.”, did you select homozygous plants in M1 or M2, how did you select? Please add this part in detail in Materials and Methods and Results.
2. In 2.4, authors mentioned that “produced a double mutant, dep1 osggc2, by crossing the mutants to investigate this hypothesis.”, please add this part in detail in Materials and Methods and Results.
3. How about the agricultural traits of single mutant dep1, osggc2, and double mutant, dep1 osggc2?
4. References No. 16: Add a dot at the end of the title sentence.
Author Response
The paper “OsGGC2, Gγ subunit of heterotrimeric G protein, regulates plant height by functionally overlapping with DEP1 in rice” reported the genetic mechanisms of Gγ subunit of heterotrimeric G protein, OsGGC2, for plant height. Authors generated loss-of-function mutants of DEP1 and OsGGC2 in rice and found that both dep1 and osggc2 mutants showed dwarf phenotype and double mutants dep1 osggc2 also showed a synergistic phenotype. Authors concluded that both DEP1 and OsGGC2 are positive regulators of plant height, and their functions are redundant.
This is a technically competent study that provides strong evidence for two type C Gγ subunits, DEP1 and OsGGC2, are positive regulators and functionally redundant for plant height in rice.
However, there are some points to check and revise before publishing.
- In 4.2, authors mentioned that “M1 seeds were genotyped by sequencing, and homozygous plants were selected.”, did you select homozygous plants in M1 or M2, how did you select? Please add this part in detail in Materials and Methods and Results.
Answer: Thank you for pointing this out. We have added relevant details as suggested.
- In 2.4, authors mentioned that “produced a double mutant, dep1 osggc2, by crossing the mutants to investigate this hypothesis.”, please add this part in detail in Materials and Methods and Results.
Answer: Thank you for pointing this out. We have added the suggested details.
- How about the agricultural traits of single mutant dep1, osggc2, and double mutant, dep1 osggc2?
Answer: Thank you for pointing this out. We understand that agronomic traits are important in rice research. However, in this study, we did not investigate agronomic traits because the single mutation reduced plant height and biomass, the double mutation significantly dwarfed the plants and clearly reduced the yield; in Japan, transgenic plants are grown in a limited space of a closed greenhouse.
- References No. 16: Add a dot at the end of the title sentence.
Answer: Thank you for pointing this out. We have added the dot.
Reviewer 2 Report
In this work, a study about the role of the Gg subunit of the rice G protein in regulating plant height and grain size was performed using defective mutants obtained by the CRISPR/CAS9 technology. I think that the manuscript is interesting and contributes to the knowledge on the role of this protein complex in regulating rice development.
Some minor suggestions/comments:
1.- In the abstract, I think the sentence “In type C Gg subunits…” is a bit confusing. DEP1 controls plant height but the role of these genes on plant height has not yet been clarified?
2.- In the introduction, line 29: Activated GPCRs, which function as an intrinsic GDP/GTP exchange factor … of GPCRs?
3.- In line 54, I think it would be better to indicate that Gg2 is a type B g subunit and that GS3, DEP1 and OsGGC2 are the three type C subunits described in rice.
4.- In line 88: please reference separately the figures, “The panicle length was not significantly different from that of WT (Figure 2c and f), and the grain length was slightly shorter only in dep1 (Figure 2d and g).”
5.- In line 97 there is a mistake: Student’s t-test are actually indicated in (e), (f) and (g). The same mistake is in the legend of Figure 3 (line 120) and in that of Figure 4 (line 139).
6.- In line 101 I think it will be helpful to describe double mutants as those with the d1 allele of the a subunit
7.- Lines 102 and 106 sound redundant.
8.- In line 119, Student’s t-test was performed between the wild type and mutants? The figures indicate significant differences between double mutants and d1.
9.- Why subheading the discussion section? Is it really necessary? Some sentences in this section are already written at the results section, for instance lines 150-152.
10.- How many plants from each genotype were used for phenotype analyses?
Author Response
In this work, a study about the role of the Gg subunit of the rice G protein in regulating plant height and grain size was performed using defective mutants obtained by the CRISPR/CAS9 technology. I think that the manuscript is interesting and contributes to the knowledge on the role of this protein complex in regulating rice development.
Some minor suggestions/comments:
1.- In the abstract, I think the sentence “In type C Gg subunits…” is a bit confusing. DEP1 controls plant height but the role of these genes on plant height has not yet been clarified?
Answer: Thank you for pointing this out. We have rewritten the sentence as follows:
“In type C Gγ subunits, GS3, which controls grain size, and DEP1, which controls plant height and panicle branching, and their homolog OsGGC2, which affects grain size, have been reported; however, the function of each gene, their interactions, and molecular mechanisms for the control of plant height, has not yet been clarified.”
2.- In the introduction, line 29: Activated GPCRs, which function as an intrinsic GDP/GTP exchange factor … of GPCRs?
Answer: Thank you for pointing this out. We have deleted “of GPCRs.”
3.- In line 54, I think it would be better to indicate that Gg2 is a type B g subunit and that GS3, DEP1, and OsGGC2 are the three type C subunits described in rice.
Answer: Thank you for pointing this out. We have rewritten the sentence as follows:
“Arabidopsis has two of type A and one type C, while rice has one type A (RGG1/ Gγ1), one type B (RGG2/ Gγ2), and three of type C (GS3, DEP1, and OsGGC2) of the five γ subunits [14].”
4.- In line 88: please reference separately the figures, “The panicle length was not significantly different from that of WT (Figure 2c and f), and the grain length was slightly shorter only in dep1 (Figure 2d and g).”
Answer: Thank you for pointing this out. We have separated the figure references as suggested.
5.- In line 97 there is a mistake: Student’s t-test are actually indicated in (e), (f) and (g). The same mistake is in the legend of Figure 3 (line 120) and in that of Figure 4 (line 139).
Answer: Thank you for pointing this out. We have corrected the mistakes.
6.- In line 101 I think it will be helpful to describe double mutants as those with the d1 allele of the a subunit
Answer: Thank you for pointing this out. We have rewritten the sentence as follows: “To clarify the signal transduction of heterotrimeric G protein in rice, we observed the phenotypes d1 dep1 and d1 osggc2 double mutants that were created by crossing the dep1 and osggc2 mutants with the Gα loss-of-function mutant d1.”
7.- Lines 102 and 106 sound redundant.
Answer: Thank you for pointing this out. We have rewritten the sentences as follows:
“Although both double mutants showed a dwarf phenotype similar to that of d1, the plant heights of d1 dep1 and d1 osggc2 were slightly shorter than that of d1 (Figure 3a, b, and e). Panicle lengths of d1 dep1 and d1 osggc2 were also slightly shorter than in d1 (Figure 3c and f). The grain lengths of d1 dep1 and d1 osggc2 were not significantly different from d1 (Figure 3d and g). These results suggest that d1 is almost epistatic to dep1 and osggc2 in plant height, although there was also a slight additive phenotype.”
8.- In line 119, Student’s t-test was performed between the wild type and mutants? The figures indicate significant differences between double mutants and d1.
Answer: Thank you for pointing this out. We have rewritten this part according to your comment.
- Why subheading the discussion section? Is it really necessary? Some sentences in this section are already written at the results section, for instance lines 150-152.
Answer: Thank you for pointing this out. We have removed the subheadings from the discussion and removed the repetitive parts.
10.- How many plants from each genotype were used for phenotype analyses?
Answer: Thank you for pointing this out. We have added a new section, “4.4. Phenotype evaluation,” in the Materials and Methods as follows:
“Plant height was measured from the bottom to the top of the main column, and panicle length was measured from the bottom to the top of the panicle of the main column; and the means of five plants from each genotype were compared. Grain length was measured for 15 grains per plant, and the means of five plants from each genotype were compared.”
Reviewer 3 Report
Chaya and colleagues studied the redundant functions of OsGGC2 and DEP1 in rice by using CRISPR/Cas9-mediated mutagenesis. While single gene null mutants only showed mild phenotypes, the double loss-of-function mutants exhibited extreme dwarfism and slightly shortened grain length, indicating the functional redundancy of these genes as positive regulators of plant height. Interestingly, double mutants of d1 dep1 and d1 osggc2 demonstrated only slightly shorter plant heights compared to the d1 single mutant, suggesting that DEP1 and OsGGC2 exert their regulatory effects mainly via the D1 (Gα)-dependent pathway. Overall, the presented data provide convincing evidence of the functional redundancy of DEP1 and OsGGC2.
Below are several minor issues:
- In the third paragraph in Page 2, please be cautious with gene/protein nomenclature. The gene products, i.e., proteins, should not be italicized. For instance, it is the gene product that control grain length and panicle branching and length. Therefore, GS3 should not be italicized in line 59; likewise, DEP1 in line 61 should not be italicized. Line 108: DEP1 and OsGGC2 should not be italicized. In page 5, line 123, DEP1 and OsGGC2 should not be italicized. In page 6, line 147, DEP1 and OsGGC2 should not be italicized. There are more places, so please pay attention and make corrections.
- Page 2, line 74: “incomplete” should be “truncated”
- Page 3, line 88 and Figure 2d: How many grains were measured for the statistical analysis? In the Figure legend, it says “n=5”. Does this mean that all grains from 5 different mutant plants were measured? Or are the data based on only 5 grains? The slight difference (7.43 mm vs. 7.23 mm) is less than 3%, and the error bars (SD) do not seem to suggest that the difference would be statistically significant. Please provide more information and confirm that if there is sufficient data to support that DEP1 controls grain length.
- Figure 1. “Cys rich” should be “Cys-rich”. Do black highlighted sequences indicate unchanged amino acids? Please add a description in the figure legend for (c) and (d).
- Figures 2-4 bar charts: Remove the horizontal lines or rearrange the numbers and labels on top of the bars to maximize the visibility.
Author Response
Chaya and colleagues studied the redundant functions of OsGGC2 and DEP1 in rice by using CRISPR/Cas9-mediated mutagenesis. While single gene null mutants only showed mild phenotypes, the double loss-of-function mutants exhibited extreme dwarfism and slightly shortened grain length, indicating the functional redundancy of these genes as positive regulators of plant height. Interestingly, double mutants of d1 dep1 and d1 osggc2 demonstrated only slightly shorter plant heights compared to the d1 single mutant, suggesting that DEP1 and OsGGC2 exert their regulatory effects mainly via the D1 (Gα)-dependent pathway. Overall, the presented data provide convincing evidence of the functional redundancy of DEP1 and OsGGC2.
Below are several minor issues:
- In the third paragraph in Page 2, please be cautious with gene/protein nomenclature. The gene products, i.e., proteins, should not be italicized. For instance, it is the gene product that control grain length and panicle branching and length. Therefore, GS3 should not be italicized in line 59; likewise, DEP1 in line 61 should not be italicized. Line 108: DEP1 and OsGGC2 should not be italicized. In page 5, line 123, DEP1 and OsGGC2 should not be italicized. In page 6, line 147, DEP1 and OsGGC2 should not be italicized. There are more places, so please pay attention and make corrections.
Answer: Thank you for pointing this out. We have carefully rechecked the text and corrected it where needed.
- Page 2, line 74: “incomplete” should be “truncated”
Answer: Thank you for pointing this out. We have rewritten it.
- Page 3, line 88, and Figure 2d: How many grains were measured for the statistical analysis? In the Figure legend, it says “n=5”. Does this mean that all grains from 5 different mutant plants were measured? Or are the data based on only 5 grains? The slight difference (7.43 mm vs. 7.23 mm) is less than 3%, and the error bars (SD) do not seem to suggest that the difference would be statistically significant. Please provide more information and confirm that if there is sufficient data to support that DEP1 controls grain length.
Answer: Thank you for pointing this out. We have added a new section, “4.4. Phenotype evaluation,” in the Materials and Methods as follows:
“Plant height was measured from the bottom to the top of the main column, and panicle length was measured from the bottom to the top of the panicle of the main column; and the means of five plants from each genotype were compared. Grain length was measured for 15 grains per plant, and the means of five plants from each genotype were compared.”
- Figure 1. “Cys rich” should be “Cys-rich”. Do black highlighted sequences indicate unchanged amino acids? Please add a description in the figure legend for (c) and (d).
Answer: Thank you for pointing this out. We have corrected the figure and the legend.
- Figures 2-4 bar charts: Remove the horizontal lines or rearrange the numbers and labels on top of the bars to maximize the visibility.
Answer: Thank you for pointing this out. We have removed the horizonal lines.